# Adjuvant Immunotherapy in Stage IIB/IIC Melanoma: Current Evidence and Future Directions

**DOI:** 10.3390/biomedicines13081894

**Published:** 2025-08-04

**Authors:** Ivana Prkačin, Ana Brkić, Nives Pondeljak, Mislav Mokos, Klara Gaćina, Mirna Šitum

**Affiliations:** 1Department of Dermatology and Venereology, Sestre Milosrdnice University Hospital Center, 10000 Zagreb, Croatia; brkic.ana1311@gmail.com (A.B.); mislavmokos50@gmail.com (M.M.); klaragacina@gmail.com (K.G.); mirna.situm@kbcsm.hr (M.Š.); 2Department for Human Sciences, School of Medicine, University of Split, 21000 Split, Croatia; 3Dermatovenereology Department, General Hospital Sisak, 44000 Sisak, Croatia; nives.pondeljak@gmail.com; 4Department for Dermatovenereology, School of Dental Medicine, University of Zagreb, 10000 Zagreb, Croatia; 5Croatian Academy of Sciences and Arts, 10000 Zagreb, Croatia

**Keywords:** adjuvant therapy, biomarkers, melanoma, immune-related adverse events, immunotherapy, PD-1 inhibitors

## Abstract

**Background**: Patients with resected stage IIB and IIC melanoma are at high risk of recurrence and distant metastasis, despite surgical treatment. The recent emergence of immune checkpoint inhibitors (ICIs) has led to their evaluation in the adjuvant setting for early-stage disease. This review aims to synthesize current evidence regarding adjuvant immunotherapy for stage IIB/IIC melanoma, explore emerging strategies, and highlight key challenges and future directions. **Methods**: We conducted a comprehensive literature review of randomized clinical trials, observational studies, and relevant mechanistic and biomarker research on adjuvant therapy in stage IIB/IIC melanoma. Particular focus was placed on pivotal trials evaluating PD-1 inhibitors (KEYNOTE-716 and CheckMate 76K), novel vaccine and targeted therapy trials, mechanisms of resistance, immune-related toxicity, and biomarker development. **Results**: KEYNOTE-716 and CheckMate 76K demonstrated significant improvements in recurrence-free survival (RFS) and distant metastasis-free survival (DMFS) with pembrolizumab and nivolumab, respectively, compared to placebo. However, no definitive overall survival benefit has yet been shown. Adjuvant immunotherapy is linked to immune-related adverse events, including permanent endocrinopathies. Emerging personalized approaches, such as circulating tumor DNA monitoring and gene expression profiling, may enhance patient selection, but remain investigational. **Conclusions**: Adjuvant PD-1 blockade offers clear RFS benefits in high-risk stage II melanoma, but optimal patient selection remains challenging, due to uncertain overall survival benefit and toxicity concerns. Future trials should integrate biomarker-driven approaches to refine therapeutic decisions and minimize overtreatment.

## 1. Introduction

### 1.1. Epidemiology and Genetic Profile of Melanoma

Melanoma is a tumor caused by the genetic transformation and subsequent hyperproliferation of melanocytes. It presents a significant public health concern and economic burden because it is the leading cause of death from skin cancers. In 2020 alone, over 325,000 new cases were diagnosed, and melanoma accounted for more than 57,000 deaths worldwide [1,2,3]. Its mortality rates are expected to double by 2040 [4].

Cutaneous melanomas show a high tumor mutational burden (TMB), particularly those arising in chronically sun-exposed regions such as the head, neck, and upper limbs [5,6,7]. Genetic studies have identified common mutations in melanoma, including cyclin-dependent kinase inhibitor 2A (CDKN2A), neuroblastoma rat sarcoma viral oncogene homolog (NRAS), phosphatase and tensin homolog (PTEN), and BRAF, with the latter being the most frequent oncogenic mutation. Additionally, neurofibromin 1 (NF1) mutations have emerged as another distinct melanoma subtype [5]. As shown in Hayward et al. [7], the median somatic mutation burden in cutaneous melanomas exceeds 100 mutations per megabase (mut/Mb), while acral and mucosal subtypes demonstrate significantly lower TMBs, with median values of approximately 10 mut/Mb and two mut/Mb, respectively [7]. Similarly, the Cancer Genome Atlas (TCGA) classification highlights that triple wild-type melanomas, common in acral and mucosal locations, show substantially reduced UV mutational signatures and lower mutation rates compared to B-RAF Proto-Oncogene, Serine/Threonine Kinase (BRAF-), neuroblastoma RAS viral oncogene homolog (NRAS-), and neurofibromin 1 (NF1) mutated subtypes [6].

This variability affects treatment responses, particularly to therapies targeting the mitogen-activated protein kinase (MAPK) pathway (B-Raf proto-oncogene, serine/threonine kinase (BRAF) and mitogen-activated protein kinase kinase inhibitor (MEK or MAPKK inhibitors)) and immunotherapy [5].

### 1.2. Rationale for Adjuvant Therapy in High-Risk Melanoma

While early-stage (stage I–IIA) melanoma can often be cured by surgical excision alone, patients with stages IIB–IIC and stage III melanoma face high recurrence rates within five years after complete resection [8,9,10]. Recent advances in melanoma treatment have significantly improved patient survival rates. Given the proven effectiveness of immunotherapy and targeted therapies in treating metastatic melanoma, it was a logical next step to explore their use in the adjuvant setting, aiming to help patients who are at high risk of recurrence or death from melanoma [5,11,12,13]. The five-year melanoma-specific survival rates are 87% for stage IIB, 82% for stage IIC, 93% for stage IIIA, 83% for stage IIIB, 69% for stage IIIC, and 32% for stage IIID, indicating a gradual decline in survival as the disease progresses to more advanced stages [14]. Outcomes among patients with stage II melanoma are, therefore, diverse, ranging from a low mortality risk in stage IIA to significantly worse outcomes in stage IIC, which has a poorer prognosis than stage IIIA and is comparable to stage IIIB [14,15]. Also, although patients with thin melanoma have a much lower risk of death compared to those with thick melanoma, the far greater number of thin melanoma cases results in a higher total number of deaths in this group, which highlights a clear and persistent unmet need for improved treatment options in earlier-stage melanoma [13].

## 2. Immune Checkpoint Inhibitors in the Adjuvant Setting

### 2.1. Mechanisms of Action

Numerous adjuvant therapies have gained approval and are now recognized as the standard of care for patients at high risk of recurrence [16]. Cytotoxic T lymphocyte-associated protein 4 (CTLA-4) is an immune checkpoint protein that regulates immune tolerance. It is expressed on T-cells and inhibits their function by interacting with B7, thereby preventing autoimmune reactivity [17,18]. Ipilimumab, an anti-CTLA-4 monoclonal antibody, blocks this inhibition, promoting T-cell activation and enhancing cancer cell recognition [18]. PD-1 is another immune checkpoint protein expressed on immune cells (T and B lymphocytes, natural killer cells, monocytes, and macrophages), which interacts with PD-L1 on tumor cells, facilitating immune evasion. It plays a pivotal role in programmed cell-death signaling and modulates T-cell-mediated immune responses. Upon binding to its ligands, PD-1 disrupts essential downstream signaling pathways necessary for T-cell activation and impairs transcriptional activity, ultimately leading to the suppression of T-cell-driven immunity. Blocking this pathway with monoclonal antibodies, such as pembrolizumab and nivolumab, restores immune activity [17,18,19,20]. More recently, inhibitors of lymphocyte-activation gene 3 (LAG-3), expressed on activated CD4+ and CD8+ T lymphocytes, have been developed [16,19,20]. Several randomized controlled trials of adjuvant immune checkpoint inhibitor therapies have been completed, with two focusing on adjuvant therapy in melanoma stages IIB and IIC (Table 1).

Although individual risk in stage II melanoma is generally low to moderate, the high incidence of early-stage diagnoses indicates that these patients contribute to approximately 30–50% of all melanoma-related deaths. Consequently, adjuvant treatments such as pembrolizumab and nivolumab have been investigated for resected stage IIB/C melanoma [14,29].

### 2.2. Pembrolizumab in Stage IIB/IIC Melanoma

The effectiveness of pembrolizumab in the treatment of stage IIB and IIC melanoma has been evaluated in a multicenter, randomized clinical trial called KEYNOTE-716. The study included 976 patients with resected stage IIB or IIC melanoma who received pembrolizumab at a dosage of 2 mg/kg (up to 200 mg) intravenously every three weeks for one year, or placebo. In its final analysis, the KEYNOTE-716 demonstrated a statistically significant enhancement in both recurrence-free survival (RFS) and distant metastasis-free survival (DMFS). The estimated 36-month RFS for stage IIB/IIC melanoma was 76.2% with pembrolizumab and 63.4% with placebo, corresponding to a hazard ratio (HR) of 0.62 (95% CI: 0.49–0.79), which reflects a 12.8% absolute improvement in recurrence-free survival. The estimated 36-month RFS was 79.7% for pembrolizumab compared to 66.5% for placebo in patients with stage IIB melanoma (HR, 0.58; 95% CI, 0.43 to 0.79) and 71.4% for pembrolizumab versus 58.0% for placebo in stage IIC melanoma (HR, 0.65; 95% CI, 0.45 to 0.94). Notably, the benefit was consistent across both stage IIB and IIC subgroups, with a slightly greater relative risk reduction in stage IIB compared to IIC, indicating meaningful efficacy in early high-risk subgroups. However, the absolute benefit was higher in stage IIC (13.4%), due to its worse baseline prognosis, underscoring the importance of considering both absolute and relative benefits when evaluating treatment impact. These findings suggest that while both subgroups derive meaningful benefit, patients with stage IIC may gain more in absolute terms. A subgroup analysis revealed that pembrolizumab significantly improved RFS across all major patient groups, including both genders and age categories (<65 and ≥65 years). The estimated 36-month DMFS rate for stage IIB/IIC melanoma patients was 84.4% for patients treated with pembrolizumab, compared to 74.7% for those who received placebo, corresponding to an HR of 0.59 (95% CI: 0.44–0.79), indicating a 9.7% absolute reduction in the risk of distant metastasis at 3 years. Among patients with stage IIB melanoma, the median DMFS remained unreached in both arms, with 36-month rates of 86.7% for pembrolizumab and 78.9% for placebo (HR: 0.62; 95% CI: 0.42–0.92). Similarly, for those with stage IIC disease, the median DMFS was not reached, and the 36-month rates were 80.9% and 68.1% for the pembrolizumab and placebo groups, respectively (HR: 0.57; 95% CI: 0.36–0.88), corresponding to a 12.8% absolute benefit. These data demonstrate that the risk of developing distant metastasis, a strong predictor of mortality, is significantly delayed in both stage IIB and IIC patients, with slightly greater absolute benefit in stage IIC. This underscores the importance of pembrolizumab as an adjuvant therapy, not only for preventing recurrence, but also for delaying life-threatening metastatic spread. In terms of safety, pembrolizumab treatment was associated with a higher incidence of treatment-related adverse events (82.6%) compared to placebo (63.6%). Still, these were of manageable severity, and no treatment-related deaths occurred. The most common treatment-related adverse events were mild to moderate (grade 1–2), including pruritus, fatigue, diarrhea, rash, arthralgia, and hypothyroidism. These results led to regulatory approval from the U.S. Food and Drug Administration (FDA) and the European Medicines Agency (EMA) for the adjuvant treatment of adult and pediatric patients (ages greater than 12) with stage IIB and IIC melanoma following complete resection [15,21,24,27,30,31,32,33,34,35].

### 2.3. Nivolumab in Stage IIB/IIC Melanoma

Two years later, after completing the CheckMate 76K trial, nivolumab became the second PD-1 inhibitor to receive approval for the same indications as pembrolizumab. In this study, 790 patients with resected stage IIB/C melanoma were randomized in a 2:1 ratio to receive either nivolumab 480 mg intravenously every four weeks for one year or a placebo. At the first interim analysis, nivolumab showed a substantial and clinically relevant improvement in RFS compared to placebo. In patients with stage IIB melanoma, the 12-month RFS rate was 92.6% with nivolumab (95% CI: 88.6–95.2) versus 84.1% with placebo (95% CI: 76.8–89.3), with a hazard ratio (HR) of 0.34 (95% CI: 0.20–0.56), and an absolute difference of 8.5%. In patients with stage IIC melanoma, the 12-month RFS was 83.8% for nivolumab (95% CI: 77.5–88.4) versus 72% for placebo (95% CI: 61.6–80.0), with an HR of 0.51 (95% CI: 0.32–0.81), and 11.8% absolute improvement. These differences indicate a meaningful delay in recurrence across both subgroups, with numerically greater absolute benefit in patients with worse baseline prognosis (i.e., stage IIC). Subgroup analysis confirmed that nivolumab significantly improved RFS across major demographic groups, including both genders and patients younger and older than 65. The safety profile of nivolumab was similar to that of pembrolizumab, with 10.3% of nivolumab-treated patients experiencing grade 3–4 treatment-related adverse events at the first interim analysis, in contrast to 2.3% in the placebo group. The most common adverse events, similar to those seen with pembrolizumab, included fatigue, pruritus, diarrhea, rash, hypothyroidism, and arthralgia. There was one treatment-related death among the 526 patients in the nivolumab group, underscoring the need for ongoing vigilance, despite overall acceptable tolerability. One-year DMFS was 92% with nivolumab versus 87% with placebo (HR 0.47 [0.30–0.72]), and a 5% absolute reduction in risk of distant metastasis. These results ultimately led to the FDA approving nivolumab as an alternative PD-1 inhibitor for adjuvant treatment of completely resected high-risk stage II melanoma. After a three-year follow-up, the CheckMate 76K study demonstrated that adjuvant nivolumab provided a significant benefit in extending both RFS and DMFS among patients with resected stage IIB/C melanoma. The 3-year RFS rate was 71% in the nivolumab group compared to 61% in the placebo group, with a hazard ratio (HR) of 0.62 and a 95% confidence interval (CI) of 0.47 to 0.80. While the difference in 3-year DMFS was numerically modest (5%), the sustained hazard ratio underlines the long-term potential of nivolumab in delaying recurrence and distant progression. Similarly, the 3-year DMFS rate was 79% with nivolumab versus 74% with placebo, with a hazard ratio of 0.72 (95% CI, 0.52–1.0), reflecting a reduced risk of distant metastasis. These results confirmed the durable efficacy of nivolumab as an adjuvant therapy in this patient population [9,15,29,40].

### 2.4. Historical Comparators and the Ongoing Challenge of Demonstrating Overall Survival Benefit

Clinical trial data show that both nivolumab and pembrolizumab significantly enhance RFS and DMFS compared to placebo or older standards, such as high-dose interferon (HDI) and ipilimumab. Regarding overall survival (OS), ipilimumab was the first to show a survival benefit in the adjuvant setting; however, it is associated with high toxicity. While anti-PD1 agents in adjuvant therapy have shown consistently positive results for RFS and DMFS, statistically significant improvements in OS are lacking, likely due to post-relapse treatment, insufficient follow-up, or immunological mechanisms that remain to be fully understood. 

In CheckMate-238, after 7 years of follow-up, no significant OS benefit was observed for nivolumab compared to ipilimumab, despite improvements in RFS and disease-free survival (DFS). This lack of OS benefit is possibly influenced by the extensive use of subsequent immunotherapy and targeted therapies in both groups, which may lead to potential confounding effects [16,21,22,26,27,41].

Similarly, the SWOG S1404 trial demonstrated a statistically significant improvement in RFS for pembrolizumab compared to high-dose interferon-α2b (IFN-α2b) or ipilimumab, but no OS benefit was observed. This multicenter study involving 1301 randomized patients provides significant evidence supporting the decision to avoid using ipilimumab and IFN-α2b as adjuvant therapies. These older treatments are less effective in preventing recurrence compared to pembrolizumab, and carry a higher risk of severe toxicity. However, it remains uncertain whether adjuvant PD-1 blockade provides an OS benefit. At a median follow-up of 47.5 months, pembrolizumab demonstrated significantly improved recurrence-free survival (RFS) compared to previous standard adjuvant immunotherapies, with a hazard ratio of 0.77 (99.62% CI: 0.59–0.99; *p* = 0.002). However, the difference in OS between treatment groups was not statistically significant (HR 0.82; 96.3% CI: 0.61–1.09; *p* = 0.15). Grade 3-to-5 treatment-related adverse events occurred in 19.5% of patients receiving pembrolizumab, compared to 71.2% with interferon alfa-2b and 49.2% with ipilimumab. As noted by the authors, demonstrating an OS benefit with adjuvant melanoma therapy has become increasingly challenging, due to significant improvements in post-recurrence survival. In this trial, pembrolizumab was associated with an 18% improvement in overall survival; however, this difference did not reach the predefined threshold for statistical significance, as the analysis was conducted when only 57% of the planned survival events had occurred. Both groups experienced longer-than-expected survival, making it challenging to detect a statistically significant difference in overall survival between treatment strategies. Additionally, the trial included patients with stage IIIA-C melanoma using the *AJCC 7th edition*, which does not directly correspond to the current *AJCC 8th edition* staging. Notably, patients classified as stage IIIA in the *8th edition* typically have a lower risk than those classified as stage IIIA in the *7th edition* [10,21,24,27,37,42].

The IMMUNED trial also found no OS benefit for adjuvant nivolumab [38]. Therefore, there are ongoing questions regarding the correlation between RFS, DMFS, and OS in clinical practice.

While KEYNOTE-716 and CheckMate 76K are landmark studies in the adjuvant treatment of stage IIB/IIC melanoma, their results must be interpreted within the broader historical and therapeutic context. Historically, high-dose interferon-α2b was the first approved adjuvant therapy. Still, it provided only modest improvements in relapse-free survival (RFS) and no consistent overall survival (OS) benefit, while causing significant toxicity in more than half of the patients [10,21,22,24,25,37]. Similarly, adjuvant ipilimumab demonstrated an overall survival benefit at the high dose of 10 mg/kg in the EORTC 18071 trial, but this was associated with a high rate of severe immune-related adverse events and significant treatment-related mortality [10,13,15,21,22,23,24]. These limitations led to its limited use in clinical practice.

While both KEYNOTE-716 and CheckMate 76K have established PD-1 inhibitors as standard options in the adjuvant setting for stage IIB/IIC melanoma based on RFS and DMFS improvements, their findings must be interpreted with caution. The absence of overall survival benefit so far, along with the use of placebo comparators and the rise of promising alternative strategies, including targeted therapies, vaccines, and biomarker-guided approaches, underscores the need for a more nuanced and personalized treatment approach in early-stage melanoma.

### 2.5. Resistance to Adjuvant Immunotherapy and Its Mechanisms

One of the potential problems in treating melanoma patients is resistance to immune checkpoint inhibitors (ICIs). Stage IIB and IIC melanomas have the highest likelihood of recurrence within the first three years after diagnosis [43]. Patients who initially present with stage IIB or IIC melanoma are more likely to experience recurrence in regional lymph nodes, followed by the lungs and in-transit locations [44]. Resistance to adjuvant anti-PD-1 therapy in melanoma refers to disease recurrence following surgical resection in patients who had no detectable metastases at treatment initiation. According to consensus definitions, primary resistance in the adjuvant setting occurs when progression develops within 12 weeks of the last dose, whereas secondary resistance refers to relapse occurring after 12 weeks [45].

Primary resistance in melanoma immunotherapy refers to an innate lack of response to treatment, which is present before therapy is initiated. The biological basis for this resistance includes tumors that are inherently less immunogenic; they either do not present enough tumor-associated antigens on MHC molecules or fail to attract sufficient cytotoxic T lymphocytes (CTLs) into the tumor microenvironment. As a result, these tumors are characterized by low T-cell infiltration and a paucity of immune activation. Additionally, a tumor’s microenvironment may contain high levels of immunosuppressive cytokines such as TGF-β and IL-10, or cells like regulatory T-cells and myeloid-derived suppressor cells, which inhibit effective immune responses. Specific signaling pathway abnormalities, such as activation of WNT/β-catenin signaling, can also prevent T-cell priming and infiltration, further contributing to primary resistance. Therapeutically, these tumors tend to respond poorly to checkpoint inhibitors such as anti-PD-1 or anti-CTLA-4 therapies. As a result, combination strategies aimed at increasing tumor immunogenicity—such as epigenetic modifiers, oncolytic viruses, radiation, or therapies that promote T-cell infiltration—are often employed to overcome this resistance [45].

DNA methylation and other epigenetic changes are common in melanoma, and usually result in the downregulation of tumor suppressor genes and decreased immune antigen presentation. These epigenetic changes can be detected through a liquid biopsy, which is, therefore, a non-invasive method for monitoring tumor activity. Epigenetic therapies that combine DNA demethylation agents with histone deacetylase (HDAC) inhibitors have demonstrated effectiveness in preclinical studies of non-small-cell lung cancer (NSCLC). Specifically, these treatments reversed MYC-driven proliferation and immune evasion, reprogramming exhausted T-cells into a more potent effector and memory phenotype, thus significantly improving the outcomes of immune checkpoint therapy [46]. HDAC inhibitors influence chromatin structure and gene expression, leading to several effects that make tumors more susceptible to immune attack. They increase the expression of tumor-associated antigens and enhance the presentation machinery by upregulating MHC class I and II molecules, which helps immune cells recognize and target cancer cells more effectively [47]. Moreover, HDAC inhibitors can modulate immune checkpoint molecules such as PD-L1, typically by downregulating their expression, thereby reducing tumor-mediated immunosuppression. These agents also favorably alter the tumor microenvironment by promoting the infiltration of cytotoxic T lymphocytes while decreasing immunosuppressive cells like regulatory T-cells and myeloid-derived suppressor cells. They also stimulate the production of pro-inflammatory cytokines, further supporting an active anti-tumor immune response [47]. The rationale behind combining HDAC inhibitors with checkpoint inhibitors is based on their complementary effects. While HDAC inhibitors prime the tumor and its microenvironment to be more immunogenic, checkpoint inhibitors release the brakes on immune cells, allowing a more effective immune response. This synergy aims to convert tumors that lack immune cell infiltration and are resistant to immunotherapy into tumors that are actively engaged by the immune system. Epigenetic modulation via HDAC inhibitors enhances tumor visibility and immune system engagement, and, when paired with checkpoint blockade, can significantly improve responses in cases where tumors initially resist immunotherapy.

In melanoma patients, innovative strategies such as fecal microbiota transplantation (FMT) combined with anti-PD-1 therapy have also been shown to be useful in overcoming resistance by modifying the gut microbiome, resulting in enhanced CD8+ T-cell activation within the tumor microenvironment. Studies indicate that melanoma patients who respond to anti-PD-1 therapy have a distinct gut microbiome composition compared to those who do not respond [48]. Moreover, personalized neoantigen-based therapies, such as individualized mRNA vaccines combined with pembrolizumab, have emerged as effective strategies against resistance. In a recent phase 2b clinical study (KEYNOTE-942), individualized neoantigen therapy (mRNA-4157/V940) combined with pembrolizumab significantly enhanced outcomes compared to pembrolizumab alone in resected melanoma patients, underscoring the potential of tailored combination strategies [39]. Also, talimogene laherparepvec (T-VEC) is an oncolytic herpes simplex virus genetically modified to express granulocyte-macrophage colony-stimulating factor (GM-CSF). It is utilized as intra-lesional therapy for unresectable melanoma, promoting both local and systemic anti-tumor responses. When combined with immune checkpoint inhibitors (ICIs) like pembrolizumab, T-VEC may exhibit synergistic effects; however, further studies are needed to confirm its clinical utility [3,5,16,18,20,49].

In contrast, acquired resistance develops after an initial positive response to immunotherapy. This form of resistance is associated with tumor evolution and adaptation during treatment exposure. Tumor cells can acquire genetic or epigenetic changes that enable immune escape. For example, they can lose or downregulate MHC class I molecules or components like β2-microglobulin, impairing antigen presentation and recognition by T-cells. Additionally, tumor cells may upregulate alternative immune checkpoints such as TIM-3 or LAG-3, which act as escape pathways even when PD-1/1/PD-L1 pathways are blocked. Activation of compensatory signaling pathways, such as reactivation of the MAPK pathway, can promote tumor survival, despite ongoing immune attack. The tumor microenvironment may also adapt over time, recruiting more immunosuppressive cells or cytokines, thereby dampening immune responses. This evolution is often driven by tumor heterogeneity, allowing resistant clones to emerge and proliferate. Clinically, these tumors initially respond to therapy, but eventually progress as the tumor cells employ various escape mechanisms. Overcoming acquired resistance generally involves using combination or sequential therapies targeting multiple immune checkpoints or pathways, or adding agents like epigenetic modulators to re-sensitize the tumor to immune attack [45].

At the molecular level, acquired mutations that affect interferon signaling or antigen presentation pathways (e.g., loss of β2-microglobulin) can impair immune recognition, facilitating immune escape. Additionally, the loss of phosphatase and tensin homolog (PTEN), a tumor suppressor, contributes to an immunosuppressive environment by activating the phosphatidylinositol 3-kinase/protein kinase B (PI3K-AKT) pathway [45,50]. On the cellular side, T-cell exhaustion, characterized by persistent expression of inhibitory receptors (Programmed Death-1 (PD-1), T-cell immunoglobulin and mucin-3 (TIM-3)), T-cell immunoreceptor with Ig and immunoreceptor tyrosine-based inhibitory motif (ITIM) domains (TIGIT), and lymphocyte-activation gene 3 (LAG-3), significantly limits durable responses [45]. The absence of memory T-cells further undermines long-term immunity after initial tumor clearance. Furthermore, the tumor microenvironment (TME) in relapsing cases often exhibits increased infiltration of regulatory T-cells (Tregs) and myeloid-derived suppressor cells (MDSCs), both of which inhibit effector T-cell function through the release of immunosuppressive cytokines (IL-10, TGF-β) and metabolic competition. Lastly, tumors may adapt through clonal evolution, losing immunogenic neoantigens over time, thereby evading T-cell detection even after a previous immune response was mounted during adjuvant therapy [45]. Recent studies have identified promising therapeutic strategies explicitly aimed at overcoming resistance mechanisms. PTEN loss, which activates the PI3K-AKT signaling pathway, significantly contributes to immune evasion by reducing tumor infiltration and the cytotoxic activity of T-cells. In preclinical melanoma models, selective PI3K-β inhibitors, such as GSK2636771, have shown potential benefits by boosting T-cell-mediated anti-tumor responses without notably impairing overall T-cell function [50]. Furthermore, the combination of relatlimab (anti-LAG-3) and nivolumab (anti-PD-1) in advanced melanoma showed improved progression-free survival compared to nivolumab alone, directly addressing T-cell exhaustion through dual immune checkpoint inhibition [51]. Similarly, combination therapy with nivolumab and ipilimumab (anti-CTLA-4) significantly enhances overall survival, highlighting its effectiveness against resistance mechanisms in advanced melanoma [52]. To overcome resistance, researchers have also been investigating several other innovative immunomodulatory strategies, such as toll-like receptor (TLR) agonists and stimulators of interferon gene (STING) agonists. Among these, TLR-9 and TLR-7/8 agonists are being studied for their ability to stimulate innate immunity by recognizing molecular motifs common to pathogenic organisms, leading to the production of type I interferon and enhanced T-cell anti-tumor response. Adoptive cell therapy (ACT) involves extracting tumor-infiltrating lymphocytes (TILs), expanding them in the laboratory, and reinfusing them. This approach aims to enhance the anti-tumor immune response, particularly in patients who do not respond to immune checkpoint inhibitors (ICIs) [3].

When planning subsequent treatment, clinicians should consider the location and timing of recurrence, as these factors can lead to the most appropriate therapeutic strategy. Patients whose disease progresses during or right after adjuvant therapy may benefit from switching to a new regimen, using a different class of drugs. Conversely, patients who achieve disease control (complete or partial response or stable disease) without significant long-term adverse effects and then experience recurrence more than three months after stopping treatment may benefit the most by resuming the same agent or class upon recurrence [15]. Moreover, concerns exist about the effectiveness of subsequent therapies following adjuvant immunotherapy. Retrospective data showed that progression post–PD–1/L1 is associated with a lower response to subsequent immunotherapy, particularly with monotherapy. However, combination strategies, such as pembrolizumab plus low-dose ipilimumab, show moderate activity and manageable toxicity, making them promising second-line options. It remains unclear whether this is due to inherited resistance or resistance acquired from prior therapy [12,53,54].

Another important factor for drug bioavailability enhancement and effectiveness is advancements in delivery systems for melanoma immunotherapy. Nanoparticle-based delivery systems, such as liposomes, polymers, dendritic polymers, inorganic nanoparticles, and exosomes, can provide controlled release and delivery with improved bioavailability. Cell-based delivery systems, such as dendritic cells (DCs) and engineered cells, enhance biocompatibility and enable targeted delivery to tumor cells. DC-based therapies activate strong T-cell responses, while engineered platelets and mesenchymal stem cells (MSCs) transport anticancer agents directly to tumors [3].

These recent advances highlight the potential and importance of multimodal therapeutic strategies that target the full range of immune resistance in melanoma. By addressing these diverse yet connected pathways, these strategies provide a rational and integrated approach to overcoming both innate and acquired resistance to immunotherapy and enhancing sustained treatment responses.

### 2.6. Immune-Related Adverse Events, Long-Term Toxicity in the Adjuvant Setting

Checkpoint inhibitors can disrupt immune tolerance and cause immune-related adverse events (irAEs) that resemble autoimmune conditions. The severity of irAEs is categorized using the Common Terminology Criteria for Adverse Events (CTCAEs), ranging from mild (grade 1) to fatal (grade 5) [18].

Immune checkpoint inhibitor (ICI) therapy, particularly the combined use of anti-PD-1 (e.g., nivolumab) and anti-CTLA-4 (e.g., ipilimumab) antibodies, is associated with shifting the T-cell phenotype from an exhausted state (high expression of inhibitory receptors such as protein PD-1, lymphocyte-activation gene 3 (Lag3+), and T-cell immunoglobulin and mucin-domain containing-3 (Tim3+)) toward an active effector state (high expression of PD-1+, intermediate expression of Lag3, and high expression of Tim3+). ICI therapy aims to reinvigorate these exhausted T-cells by blocking the inhibitory signaling pathways, thereby restoring their effector functions, which can potentially trigger widespread immune activation responsible for immune-related adverse events (irAEs). Recent studies suggest that changes in circulating T-cell populations in patients treated with ipilimumab occur before the development of irAEs. Additional research highlights distinct B-cell patterns, particularly with combination therapy (ipilimumab plus nivolumab), resulting in an overall decrease in peripheral B-cell counts, but an increase in specific subsets and plasmablast cells. These B-cell changes correlate with both the frequency and severity of irAEs, suggesting that altered B-cell dynamics during combination ICI therapy may predict an increased risk of multiorgan toxicity, detectable approximately three weeks before symptom onset [18]. Long-term hormone therapy may be required for conditions such as hypothyroidism, adrenal insufficiency, or hypogonadism, which can have lasting health impacts, especially when treating patients with early-stage disease who may have been cured by surgery alone [18].

In the KEYNOTE-054 trial, a health-related quality of life (HRQoL) substudy using the 30-item EORTC QLQ-C30 demonstrated that one year of adjuvant pembrolizumab preserved HRQoL, compared to placebo. However, a subset of patients experienced severe toxicities, some of which were permanent or life-threatening. Alongside common grade 3–4 events such as colitis, pneumonitis, rash, and hepatitis, endocrine-related toxicities were significant [13,15,22].

In the EORTC 18071 trial, the significant therapeutic benefit of ipilimumab was associated with severe immune-related adverse events (AEs). The incidence and severity of these AEs were strongly dose-dependent: high-grade (3 or 4) immune-related AEs were reported in 41.6% of patients receiving high-dose ipilimumab (10 mg/kg), compared to 2.7% in the placebo group. Notably, 1.1% of treated patients died from these AEs. To address these toxicity issues, the ECOG-1609 trial evaluated whether reducing the dose of ipilimumab to 3 mg/kg could maintain effectiveness while reducing side effects. Results indicate similar efficacy in preventing recurrence at both doses, but significantly fewer adverse events at the lower 3 mg/kg dose [10]. 

In the Checkmate 238 study, adverse event rates favored the nivolumab group, with grade 3 or 4 toxicities occurring in only 14.4% of patients, compared to 45.9% in the ipilimumab group. Additionally, fewer patients discontinued treatment due to adverse events in the nivolumab arm (9.7%) than in the ipilimumab group (42.6%). Thus, the authors concluded that adjuvant nivolumab therapy improved recurrence-free survival, while significantly reducing the incidence of severe toxicity compared to high-dose ipilimumab [10].

The incidence of adverse events associated with anti-PD-1 therapy appears to be comparable between patients receiving adjuvant treatment for stage IIB/C melanoma and those treated in the advanced stage III/IV setting. Specifically, in the CheckMate-76k study, nivolumab had comparable irAE rates for high-risk stage II melanoma compared to stage III/IV melanoma [29].

In the KEYNOTE-716 trial, pembrolizumab was associated with significantly higher rates of adverse events compared to placebo (83% vs. 53%), including more grade 3 events, such as hypertension, diarrhea, autoimmune hepatitis, and rash (17% vs. 5%). The adverse events associated with pembrolizumab are primarily immune-related, and are a result of PD-1 blocking. Serious treatment-related adverse events (most commonly adrenal insufficiency, colitis, and autoimmune hepatitis) occurred in 10% of pembrolizumab recipients versus 2% with placebo. Treatment-related endocrinopathies (24% of patients) were often permanent, requiring long-term hormone therapy for treatment-related hypothyroidism, adrenal insufficiency, thyroiditis, hypophysitis, or type 1 diabetes, in most cases. Treatment discontinuation due to side effects was more common with pembrolizumab (17% vs. 5%).

Similar toxicity trends were seen in the CheckMate-76K trial with adjuvant nivolumab. Treatment-related adverse events among patients treated with adjuvant nivolumab were 83%, compared to 54% of patients in the placebo group. There was also an increased frequency of grade 3–4 adverse events (10% vs. 2%), a higher rate of immune-mediated adverse events (41% vs. 17%), and a higher rate of treatment discontinuation due to adverse events (17% vs. 3%) [32].

Despite these toxicities, pembrolizumab was approved by the FDA in 2021 for the adjuvant treatment of stage IIB/C melanoma, due to its benefit in recurrence-free survival (RFS). However, uncertainty remains about whether treating early with immunotherapy is better than waiting until recurrence, especially since many metastatic cases are now curable, and some patients may be exposed to unnecessary toxicities.

There is limited trial data evaluating combination immunotherapy, specifically in stage IIB/IIC melanoma. In the CheckMate 915 trial, no difference in RFS was observed between the combination therapy of nivolumab and ipilimumab and nivolumab alone, in stage IIIB-IV melanoma. However, combination therapy led to significantly higher rates of adverse events [21,22,36]. Given these results and the similar toxicity profiles observed with monotherapy in stages III/IV and IIB/IIC melanoma, combination adjuvant immunotherapy involving anti-CTLA4 and anti-PD-1 therapies could cause serious adverse effects in stage II settings. Exploring combination therapies with improved toxicity profiles remains essential for future research.

Additionally, since targeted BRAF/MEK inhibitor therapies have historically demonstrated lower adverse event rates than immunotherapy, they represent an important and promising area for future exploration [12,15,22,55].

Immune-related adverse events (irAEs) remain a significant barrier to the widespread use of combination immune checkpoint inhibitor (ICI) therapy, frequently resulting in early treatment discontinuation. Therefore, there is a critical need for reliable biomarkers to predict both therapeutic outcomes and the likelihood of developing irAEs [18].

Additionally, it is important to mention that melanoma often affects young adults, including those in their childbearing years. Although immunotherapies like anti-PD–1 and targeted agents (BRAF/MEK inhibitors) differ from traditional chemotherapy, their effects on fertility remain poorly understood. Pituitary dysfunction, a known irAE of PD-1 inhibitors, can impair reproductive hormones (LH, FSH, GH, and oxytocin), affecting fertility and pregnancy outcomes. Female fertility may be impacted by disrupted hormonal support for conception and gestation, while male patients with hypogonadism may face reduced fertility due to testosterone-induced suppression of spermatogenesis. PD-1 itself may play a role in pregnancy maintenance, and PD-1 gene variants have been linked to recurrent miscarriage. Long-term fertility effect data are lacking, due to the relatively recent adoption of these treatments. Additionally, vemurafenib has been associated with intrauterine growth restriction, potentially because of the MAPK pathway’s role in placental development. The long-term impact of these therapies on gonadal function warrants further study, especially as their use expands in curative-intent settings [18].

While immune-related endocrinopathies are well documented with checkpoint inhibitors, the impact of these agents on fertility remains underexplored. PD-L1 is expressed in murine Sertoli and germ cells, suggesting a potential role in protecting the testicular environment. Clinical reports have described immune-related orchitis, hypogonadism, oligospermia, and, in rare cases, irreversible azoospermia and Sertoli cell-only syndrome following ICI therapy [56]. Preclinical and clinical evidence suggests that immune checkpoint inhibitors (ICIs) may adversely affect female fertility through both primary ovarian toxicity and secondary endocrine disruption. Hypophysitis, a known immune-related adverse event particularly associated with ipilimumab, can impair gonadotropin secretion, leading to hypogonadism and menstrual dysfunction. Additionally, animal studies indicate that ICIs may directly damage ovarian tissue by reducing follicular reserves, and trigger inflammation-mediated follicle depletion [57]. Also, preclinical studies suggest that blockade of PD-1/PD-L1 and CTLA-4 pathways interferes with fetomaternal immune tolerance, and may lead to miscarriage or stillbirth [57,58].

Given the increasing use of adjuvant immunotherapy in younger patients with high-risk stage II melanoma, fertility preservation strategies such as sperm or oocyte cryopreservation should be discussed before treatment initiation. Current guidelines recommend personalized fertility counseling and long-term endocrine follow-up [59]. More research is needed to determine the true incidence, mechanisms, and reversibility of these effects, and dedicated studies on the reproductive toxicity of ICIs are necessary.

## 3. Combination Adjuvant Therapies: Targeted Therapy and Vaccination Strategies with Immunotherapy

### 3.1. BRAF/MEK Inhibitors in the Adjuvant Setting

In addition to immunotherapy, BRAF and MEK inhibitors (dabrafenib + trametinib) are approved as adjuvant treatment for patients with stage III BRAF-mutant melanoma, based on the COMBI-AD trial [60]. The use of BRAF and MEK inhibitors is more limited, given that BRAF mutations are present in approximately 50% of patients diagnosed with melanoma [10]. Although no targeted therapies are currently approved for high-risk stage II melanoma, clinical trials are in progress. The EORTC-2139/COLUMBUS AD trial is enrolling 815 patients with resected stage IIB/IIC BRAFV600-mutant melanoma, to receive either 12 months of adjuvant encorafenib plus binimetinib or a placebo. Clinical trials such as KEYVIBE-101 (which includes stage IIB/IIC patients) and the cemiplimab + fianlimab study in resected stage IIC melanoma aim to assess the synergy between anti-PD-1 antibodies and targeted therapies. This approach builds on preclinical and metastatic data, showing that BRAF/MEK inhibitors can improve antigen presentation and reduce immunosuppressive cytokines, potentially enhancing immunotherapy response. Combination strategies may also offer a more favorable toxicity profile than dual checkpoint blockade, especially in the adjuvant setting where the risk–benefit balance is critical [12,15,22,35,55].

### 3.2. Personalized Cancer Vaccines: KEYNOTE-942 and the Promise of mRNA-4157

The KEYNOTE-942 trial is a study evaluating the effectiveness of the personalized cancer vaccine mRNA-4157 (V940) in combination with pembrolizumab, versus pembrolizumab alone, for patients with high-risk stage II-IV melanoma. The combination therapy demonstrated statistically significant and clinically relevant improvements in RFS compared to pembrolizumab alone. Most treatment-related adverse events at 18 months of follow-up were mild to moderate (grade 1 or 2). The incidence of grade 3 or higher treatment-related adverse events was comparable between the two groups (25% in the combination arm vs. 18% in the monotherapy arm), with fatigue being the most frequently reported grade 3 event associated with mRNA-4157. No grade 4 or 5 events related to mRNA-4157 occurred, and the addition of mRNA-4157 did not elevate the rate of immune-mediated adverse events. At 2.5 years, the RFS rate was also higher with the combination treatment, compared to pembrolizumab alone (74.8% versus 55.6%). The combination led to improvement in DMFS with an HR of 0.384 (95% CI: 0.172–0.858). Overall survival (OS) also favored the combination therapy, with a 2.5-year OS rate of 96.0% compared to 90.2% with pembrolizumab alone, corresponding to a hazard ratio of 0.425 (95% CI: 0.114–1.584). These emerging data could significantly impact treatment approaches for patients with high-risk stage II melanoma [15,39,61].

To synthesize and compare the most relevant adjuvant treatment strategies for stage IIB/IIC melanoma, including monotherapy and emerging combination approaches, we provide an overview in Table 2. This includes PD-1 inhibitors, BRAF/MEK-targeted combinations, and vaccine-based immunotherapy.

## 4. Biomarkers: Tools for Personalizing Adjuvant Therapy

Biomarkers for melanoma immunotherapy resistance are measurable biological indicators, such as gene expression patterns, protein levels, genetic mutations, or cellular features, which can predict whether a patient’s melanoma tumor is likely to respond or resist immune checkpoint blockade therapies. Examples include tumor mutation burden, PD-L1 expression levels, T-cell infiltration status, and gene signatures related to immune activity or suppression. The absence of immune gene signatures or T-cell infiltration before treatment can serve as predictive biomarkers of primary resistance. Biomarkers can be categorized as the following: (1) tissue-based biomarkers, such as PD-L1, TMB (tumor mutational burden), gene expression profiles (GEPs), and tumor-infiltrating lymphocytes (TILs); (2) blood-based biomarkers: circulating tumor DNA (ctDNA), immune cell subsets (e.g., CD8+ T-cells), cytokines (e.g., IL-6), and serum proteins (e.g., LDH, CRP), and (3) other biomarkers, such as gut microbiota. A multi-biomarker approach combining these categories is recommended to enhance predictive accuracy and guide individualized treatment strategies [49]. 

Current tissue-based biomarkers, such as PD-L1 and BRAF mutation status, are not yet functional in the adjuvant setting. Additionally, tumor mutational burden (TMB), which represents the total number of somatic mutations per million bases, and the reduction or loss of proteins associated with antigen presentation (such as major histocompatibility complex) can also predict resistance. Of note, cutaneous melanoma usually has a higher TMB, which makes it highly immunogenic, and an ideal candidate for immunotherapy. Furthermore, several gene expression profiling (GEP) tools, such as MelaGenix, CAM-121, Decision-Dx Melanoma, and Skyline, are currently being developed [12,15,62]. The NivoMela trial utilizes the MelaGenix 8-gene score to stratify patients with stage II melanoma. Approximately 60% of patients with stage II melanoma are classified as high-risk melanoma, and are randomized 2:1 to receive adjuvant nivolumab or a placebo [63]. Although tissue biomarkers have limited positive predictive value, they are more reliable in identifying low-risk patients (e.g., CP-GEP low-risk patients had a 5-year RFS of 93%), which helps reduce overtreatment [64].

In the case of blood-based markers, melanoma releases DNA into the bloodstream, which can be collected from peripheral blood and examined using highly sensitive methods, such as next-generation sequencing (NGS). Studies in melanoma have shown that post-surgical ctDNA detection strongly predicts recurrence and poorer overall survival. While sensitivity immediately after surgery is limited, it improves with longitudinal monitoring [12,15,65].

However, despite their promise, tissue- and blood-based biomarkers such as ctDNA and PD-L1 are limited by technical and biological challenges. The sensitivity of ctDNA detection shortly after surgery may be insufficient, due to low tumor burden, while the inconsistency in findings regarding PD-L1 as a predictive marker may result from the lack of standardized methods for determining PD-L1 thresholds across different studies [15]. These limitations make real-time treatment decisions more difficult in the adjuvant setting.

Compared to other solid tumors such as non-small-cell lung cancer, where PD-L1 status guides treatment decisions, melanoma lacks a single biomarker with validated predictive utility for patient selection [66].

Circulating tumor cells (CTCs) offer insights into the tumor of origin and metastatic sites, with studies showing that melanoma patients with PD-L1-positive CTCs are significantly more likely to respond to pembrolizumab compared to those without detectable PD-L1-positive CTCs. The predictive value of PD-L1 expression is complicated, because some patients with low PD-L1 levels can still respond to checkpoint inhibitors, while others with high levels do not. Consequently, PD-L1 expression alone is insufficient as a biomarker for determining which melanoma patients should receive anti-PD-1 therapy or whether they would benefit more from monotherapy or combination immunotherapy. The inconsistency in findings regarding PD-L1 as a predictive marker may result from the absence of standardized methods for determining PD-L1 thresholds across different studies. Additionally, PD-L1 expression is dynamic rather than fixed, due to the complex interactions between tumors and immune cells throughout treatment and can, therefore, vary depending on the timing of the biopsy. In the KEYNOTE-054 trial, patients with high-risk stage III melanoma derived benefit from pembrolizumab regardless of PD-L1 status, and this marker was not reported at all in KEYNOTE-716 [32]. Additionally, circulating tumor DNA (ctDNA) levels, which reflect cell turnover, have been linked to therapeutic outcomes, with higher baseline ctDNA associated with poorer overall survival. Circulating tumor microRNAs (miRNAs) have also been evaluated, particularly in advanced melanoma, where specific miRNAs show increased expression following immunotherapy failure [65].

Clinical implementation of biomarkers is complicated by a lack of standardization in testing methodologies and cut-offs, variable regulatory approval, and inconsistent reimbursement, which all limit their adoption in routine practice [67].

In light of the growing use of adjuvant immunotherapy, several cross-cutting challenges and enabling tools need to be addressed to optimize treatment outcomes. These are summarized in Table 3, including resistance mechanisms, strategies to overcome them, immune-related adverse events, and biomarker-guided patient selection.

## 5. Discussion

It is crucial to thoroughly evaluate the risks and benefits of immunotherapy, particularly in the adjuvant setting, where tolerance for toxicity should be lower. Survival rates indicate that some patients may experience disease recurrence or metastasis, while others can be cured solely through surgery. This raises important questions about the optimal timing for adjuvant therapy, whether it should be administered immediately after surgery or delayed until the disease recurs, and whether the potential benefits of adjuvant treatment justify the associated risks. Despite the approval of several therapies for advanced melanoma, as of May 2025, PD-1 inhibitors, specifically pembrolizumab and nivolumab, are the only class of drugs that have completed phase III randomized controlled trials and received FDA approval for the adjuvant treatment of stage IIB/IIC melanoma.

The benefits of adjuvant pembrolizumab in stage II melanoma are modest. At 3 years, it showed a 9.7% absolute reduction in distant metastasis risk [30]. Nivolumab showed similar results, with a 5% DMFS improvement at 12 months [29]. As a result, more patients may undergo treatment to prevent recurrence, increasing their risk of experiencing treatment-related toxicity. Based on recurrence-risk estimates from the German health insurance claims database and modeled hazard ratios between 0.50 and 0.75 from adjuvant immunotherapy trials, it is estimated that five to nine patients with stage IIB and four to seven with stage IIC melanoma would need to be treated to prevent one recurrence [68], with additional uncertainty regarding the benefit of adjuvant therapy in stage IIIA patients with low sentinel node tumor burden (<1 mm), particularly those with tiny deposits (<0.3 mm) [69]. These high NNT values indicate that the majority of treated patients will not experience direct benefits, highlighting the need for more targeted treatment strategies. Practical ways to lower risks could include protocols such as limiting enrollment to patients with detectable post-operative ctDNA, using risk-based treatment duration protocols (for example, 3- to 6-month PD-1 courses for ctDNA-negative or low-risk GEP patients, and reserving the complete 12-month regimen for those with high molecular risk), and ctDNA monitoring every three months to allow early discontinuation in persistently negative patients [63,64,65]. A risk-adjusted plan like this could reduce overall toxicity and costs, while maintaining benefits for the minority who truly need treatment.

Moreover, the decision to administer adjuvant immunotherapy in stage IIB and IIC melanoma patients is further complicated by the currently unconfirmed significant association between RFS and OS [11,12]. Multiple phase III trials in resectable melanoma have not shown a statistically significant overall survival (OS) benefit from adjuvant immunotherapy, despite clear improvements in recurrence-free survival (RFS). Key studies—including CheckMate-238, SWOG S1404, and IMMUNED—did not demonstrate an OS advantage. Additionally, the final OS analysis of the pivotal KEYNOTE-054 trial has been delayed until late 2027, leaving the long-term survival impact uncertain [70]. Because definitive OS evidence is lacking, current decisions can mostly depend on surrogate endpoints (RFS/DMFS). Clinicians should therefore engage in shared decision-making counseling, where adjuvant therapy for stage IIB and IIC melanoma may be postponed for patients with favorable prognostic factors or low toxicity tolerance or initiated with a clear explanation that long-term survival benefits are still hypothetical [35].

A Swedish population-based registry study, presented at the ESMO Congress in 2024, investigated the impact of adjuvant immunotherapy on overall survival (OS) in patients with stage III sentinel lymph node-positive cutaneous melanoma. The study included 1376 patients and compared outcomes between two cohorts: those diagnosed before the national implementation of adjuvant therapy (January 2016 to August 2018) and those diagnosed after its introduction (September 2018 to December 2020). After a median follow-up of three years in the post-adjuvant cohort, there was no significant improvement in melanoma-specific or overall survival [71]. These findings suggest that, in a real-world setting, the addition of adjuvant immunotherapy did not translate into a measurable survival benefit within the studied timeframe. Importantly, these real-world data mirror the lack of OS benefit reported in randomized trials CheckMate-238, SWOG S1404, and IMMUNED [26,37,38,70]. This risk–benefit imbalance, combined with high treatment costs, highlights the need to minimize overtreatment and prioritize refined patient selection.

A real-world study involving 952 patients with resected stage IIB/C melanoma reported a median overall survival (OS) of 117.6 months, with significantly longer OS in stage IIB (132.9 months) compared to IIC (104.7 months; *p* = 0.031). Five-year OS rates were 75% and 70%, respectively. Since this analysis predates the use of adjuvant PD-1 inhibitors, it provides a baseline for assessing treatment impact [72]. The study emphasizes that many patients with stage II melanoma may experience long-term survival without systemic therapy, reinforcing the importance of carefully weighing the potential benefits of adjuvant immunotherapy against the risks of overtreatment and toxicity.

However, making individualized risk–benefit decisions is challenging, due to the current lack of reliable predictive biomarkers. One potential approach to selecting candidates for adjuvant immunotherapy is to develop innovative trials that utilize molecular selection to identify populations at a higher risk of recurrence beyond pathological staging. For instance, the presence of circulating tumor DNA (ctDNA) after surgery may signal the presence of residual disease that could respond to systemic therapy, potentially enabling earlier clinical intervention. Gene expression profiling (GEP) shows promise in identifying patients at higher risk of metastasis, but its utility in guiding adjuvant therapy was not evaluated in KEYNOTE-716 or CheckMate-76K. By analyzing several genes or proteins simultaneously, GEP provides a more comprehensive understanding of the tumor microenvironment and immune dynamics. The NivoMela trial (NCT04309409) is currently prospectively testing this by using the MelaGenix GEP score to identify high-risk patients with resected stage IIA–IIC melanoma and randomizing them to receive either adjuvant nivolumab or observation. Results from this trial could help tailor treatment strategies to ensure that patients more likely to benefit from PD-1 blockade receive, it while sparing others unnecessary treatment.

Amaral et al. validated a clinicopathologic and gene expression profiling model (CP-GEP) to identify stage I/II melanoma patients at high risk of recurrence, especially those in stage I/IIA—a group currently not eligible for adjuvant therapy. Using a retrospective cohort of 543 patients with primary cutaneous melanoma, the CP-GEP model stratified patients into high- and low-risk groups, with significantly different 5-year relapse-free survival (RFS) rates (77.8% vs. 93.0%). In patients without sentinel lymph node biopsy (SLNB), the model detected six out of seven relapses, supporting its usefulness even in SLNB-negative or SLNB-omitted cases. The results show that CP-GEP outperforms AJCC staging alone, especially in early stages, and could help guide personalized treatment decisions or reduce unnecessary interventions. The study concludes that CP-GEP could complement or even replace SLNB in future clinical pathways for early-stage melanoma risk assessment [64].

These developments underscore the increasing trend toward personalized adjuvant strategies, shifting from traditional staging to utilizing molecular residual disease and gene expression-based stratifiers. However, recent consensus guidelines from the Society of Surgical Oncology concluded that gene expression profiling (GEP) tests, including 31-GEP, CP-GEP, and 11-GEP, remain investigational and are not recommended for guiding sentinel lymph node biopsy, surveillance, or adjuvant therapy decisions in primary cutaneous melanoma. While GEP may identify patients at higher risk of recurrence, current evidence does not support its routine clinical use, and further validation in prospective trials is needed [73].

Experimental strategies for overcoming immunotherapy resistance (e.g., PI3K-β and HDAC inhibitors, and FMT) have shown promise in preclinical models and early-phase clinical trials, but also remain investigational. Unlike FDA-approved adjuvant therapies (e.g., PD-1 inhibitors), these modalities require further validation before integration into routine clinical practice [46,50].

Resistance to immune checkpoint inhibitors (ICIs) remains a challenge in melanoma treatment. Predictive factors for immunotherapy response include tumor mutational burden (TMB) and loss of antigen presentation. Cutaneous melanoma usually shows high TMB, which contributes to its immunogenicity. Circulating tumor cells (CTCs), especially those expressing PD-L1, have been linked to better responses to pembrolizumab. However, PD-L1 expression alone is an unreliable biomarker, due to assay variability and its dynamic nature. In KEYNOTE-054, benefit was observed regardless of PD-L1 status, a finding not reported in KEYNOTE-716 [32].

As adjuvant options expand and demonstrate promise in high-risk stages IIB/C melanoma patients, research should focus on personalizing treatment. Trial designs need to include biomarker-based risk stratification, cost-effectiveness measures, and long-term toxicity monitoring, to maximize benefits and minimize harm.

## 6. Conclusions

Adjuvant therapy is potentially curative, and may prevent complications, as well as the poor prognosis associated with disease relapse and metastasis. Large randomized trials, such as KEYNOTE 716 and Checkmate 76K, have shown significant improvements in RFS and DMFS with pembrolizumab and nivolumab, compared to placebo. However, the absence of a demonstrated overall survival advantage in trials for high-risk stage II and III melanoma patients raises a critical debate: is it better to treat early or to delay treatment until there is potential recurrence, thus avoiding unnecessary therapy and possible treatment-related irreversible adverse effects for those who might be cured by surgery alone? Future strategies should focus on individualized risk assessment and the development of predictive biomarkers, to ensure that only patients most likely to benefit from adjuvant treatment are exposed to its risks.

## Figures and Tables

**Table 1 biomedicines-13-01894-t001:** Recent studies on the use of adjuvant immune checkpoint inhibitors in the treatment of melanoma. DMFS: distant metastasis-free survival; OS: overall survival; RFS: relapse-free survival, ↑: increase, ↓: decrease.

Trial Name	Agents Compared	Stage(s)	Main Findings	(Reference Number)
EORTC 18071	Ipilimumab vs. Placebo	III	↑ RFS and OS with ipilimumab (10 mg/kg); high toxicity	[10,13,15,21,22,23,24]
E1609	Ipilimumab (3 and 10 mg/kg) vs. Interferon	IIIB–IV	Ipilimumab 3 mg/kg improved OS over interferon; less toxic than 10 mg/kg	[10,21,22,24,25]
CheckMate 238	Nivolumab vs. Ipilimumab	IIIB–IV	Nivolumab superior in RFS, lower toxicity	[10,13,21,22,24,26,27]
EYNOTE-054	Pembrolizumab vs. Placebo	IIIA–IIIC	↑ RFS, ↑ DMFS; crossover allowed	[13,15,21,24,28]
CheckMate 76K	Nivolumab vs. Placebo	IIB–IIC	↓ risk of recurrence by 58%	[29]
KEYNOTE-716	Pembrolizumab vs. Placebo	IIB–IIC	↑ RFS and DMFS	[9,10,15,21,24,27,30,31,32,33,34,35]
CheckMate 915	Nivolumab + Ipilimumab vs. Nivolumab	IIIB–IV	No added benefit from combination, ↑ toxicity	[10,21,22,24,36]
SWOG S1404	Pembrolizumab vs. Interferon/Ipilimumab	III-IV	↑ RFS for pembrolizumab; OS trend not significant	[10,21,24,27,37]
IMMUNED	Nivolumab + Ipilimumab vs. Nivolumab vs. Placebo	IV	Combination > Nivolumab > placebo for RFS	[24,38]
KEYNOTE-942 (phase 2b)	mRNA-4157/V940 + Pembrolizumab vs. Pembrolizumab	IIIB–IV	↓ recurrence by 44% with vaccine+ pembrolizumab	[39]

**Table 2 biomedicines-13-01894-t002:** Therapeutic Strategies in Adjuvant Melanoma.

Therapeutic Strategy	Key Elements	Clinical Impact
Adjuvant Immunotherapy (PD-1 Inhibitors)	KEYNOTE-716 and CheckMate 76K demonstrated improved RFS and DMFS using pembrolizumab and nivolumab in stage IIB/IIC melanoma. Regulatory approvals granted. OS benefit unconfirmed.	New adjuvant standard of care for high-risk stage II melanoma; reduces recurrence, but long-term survival benefit remains unclear.
Immunotherapy + BRAF/MEK Inhibitors	Ongoing trials (e.g., COLUMBUS AD, KEYVIBE-101) assess synergy of PD-1 inhibitors with BRAF/MEK inhibitors to enhance antigen presentation and reduce immunosuppression in BRAF-mutant melanoma.	Potentially effective and less toxic than dual checkpoint blockade. May expand treatment to molecularly selected patients.
Immunotherapy + Personalized Neoantigen Vaccines (mRNA-4157)	KEYNOTE-942 showed that adding mRNA-4157 vaccine to pembrolizumab significantly reduced recurrence and improved RFS/DMFS. Designed to generate immune responses against patient-specific tumor neoantigens.	Represents a personalized and scalable strategy with promising efficacy and tolerability; may define future of adjuvant melanoma therapy.

**Table 3 biomedicines-13-01894-t003:** Summary of immunotherapy-associated challenges and investigational interventions.

Immunotherapy-Related Domain	Key Elements	Clinical Impact
Mechanisms of resistance	Includes tumor-intrinsic changes (e.g., PTEN loss, β2-microglobulin mutations), T-cell exhaustion, immunosuppressive microenvironment, (Tregs, myeloid-derived suppressor cells), and clonal evolution and loss of immunogenic neoantigens in tumor.	Limits durability of response. Understanding mechanisms can guide second-line treatments and patient stratification.
Investigational strategies to overcome resistance	Emerging strategies include PI3K-β and HDAC inhibitors, microbiome modulation (FMT), dual checkpoint blockade (e.g., nivolumab + relatlimab), and neoantigen vaccines; immunomodulatory strategies such as TLR and STING agonists have shown promise in preclinical models and early-phase clinical trials. These are not yet FDA-approved for adjuvant melanoma.	May re-sensitize tumors to therapy and improve outcomes in resistant melanoma. Validation in prospective trials needed.
Immune-related adverse events (irAEs)	Common irAEs include colitis, hepatitis, and endocrinopathies (often permanent). Long-term hormone therapy often required.	Warrants careful patient counseling and monitoring. May limit widespread adoption of adjuvant immunotherapy in lower-risk patients.
Biomarkers for patient selection	No single validated biomarker. They can be categorized as tissue-based biomarkers (e.g., PD-L1, TMB, GEP, TIL), blood-based biomarkers (ctDNA, immune cell subsets, cytokines, and serum proteins), and other biomarkers such as gut microbiota. ctDNA and gene expression-profiling tools (MelaGenix, CP-GEP) show promise. PD-L1 expression is inconsistent, due to biological and technical variability.	May enable precision medicine and reduce overtreatment, but real-world implementation is challenged by lack of standardization and validation.

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
