# Peer review of "Adjuvant Immunotherapy in Stage IIB/IIC Melanoma: Current Evidence and Future Directions"

_biomedicines, 2025, doi:10.3390/biomedicines13081894_

Round 1
Reviewer 1 Report
Comments and Suggestions for Authors
The manuscript provides a thorough, well-structured update on adjuvant immunotherapy for high-risk stage II melanoma, integrating pivotal trials (KEYNOTE-716, CheckMate 76K). However, it falls short in critically analyzing limitations, contextualizing findings within the broader melanoma treatment landscape, and offering actionable future directions. Structural improvements and deeper discussions on critical issues would enhance its academic rigor and clinical relevance.
1. Structural Disorganization: The manuscript's flow is disjointed (e.g., resistance mechanisms and toxicity are separated from related trial discussions), reducing readability and coherence. Sections 6 (Resistance Mechanisms) and 7 (Toxicity) should be integrated into Section 2. Section 5 (Surveillance Strategies and Recurrence Patterns) is not aligned with the core focus; it should be condensed and merged with Section 7 (Toxicity). Resistance mechanisms (e.g., T-cell exhaustion, PTEN loss) are only superficially outlined, lacking discussion of recent advances or countermeasures (e.g., novel combination therapies).
2. Insufficient Contextualization of Trial Data: Overemphasis on KEYNOTE-716/CheckMate 76K results without adequate comparison to historical treatments (e.g., interferon) or alternative strategies. Contextualizing outcomes within the broader melanoma treatment landscape is essential. Figures 1 and 2 merely display trial results without in-depth analysis or novel insights; replace with a consolidated table or remove. The absence of OS benefit is noted but not critically explored (e.g., impact of post-recurrence therapies, short follow-up).
3. Thematic Misalignment in Sections 3–4: Sections 3 (BRAF/MEK Inhibitors) and 4 (Cancer Vaccines) are not fully integrated with the adjuvant immunotherapy theme. Merge into a new section: "Combination Adjuvant Therapies," detailing immunotherapy with BRAF/MEK inhibitors or vaccines.
4. Underdeveloped Biomarker Section: Section 8 (Biomarkers) lacks depth on clinically relevant biomarkers (e.g., CTCs mentioned only in Discussion). Expand to include: a. Current limitations (sensitivity, standardization challenges); b. Comparative analysis of melanoma vs. other solid tumor biomarkers; c. Practical barriers to clinical adoption.
5. Overly Complex Discussion: Section 9 (Discussion) is overly complex and lengthy, with extraneous content. While acknowledging the need for better patient selection, no actionable strategies or ongoing research (beyond cursory mentions like NivoMela) are proposed. The high number needed to treat and costs of overtreatment warrant discussion.
Reviewer 2 Report
Comments and Suggestions for Authors
- In table 1, it is better to have a separate column for references.
- Including a schematic figure summarizing key concepts such as different therapeutic strategies, mechanisms of action, or Mechanisms of resistance to adjuvant immunotherapy—would significantly enhance the clarity and visual impact of the article.
Reviewer 3 Report
Comments and Suggestions for Authors
The present article provides a comprehensive overview of the benefits, challenges, and future directions of adjuvant therapy in early-stage high-risk melanoma. With a balanced approach, the authors effectively highlight both the therapeutic potential and the existing limitations, making this work highly valuable for researchers and clinical practitioners. However, consideration of the following points could further enhance the quality and impact of the manuscript:
- In lines 43-56: It is better to use articles with quantitative statistics about body points and mutation rate so that the results are more documented.
- Lines 175-92: It is better to do more in-depth analysis in the statistical results of trials rather than just stating the statistical data.
- It is suggested to consider a separate article for fertility and reproductive effects in the Immune-related adverse events (irAEs) section and talk about infertility separately.
- It seems that this article has used more articles that have obtained successful results, please use articles that have shown the lack of superiority of immunotherapy.
- Methods such as FMT or nanoparticles have been mostly investigated in animal models or phase I/II, but are mentioned in the discussion next to FDA approved treatments, please emphasize that these drugs are still in clinical phases.
Round 2
Reviewer 1 Report
Comments and Suggestions for Authors
The revised manuscript represents a significant improvement, demonstrating enhanced scholarly rigor and critical analysis. The authors have partially addressed prior concerns regarding structural organization, contextual depth, biomarker evaluation, and thematic coherence. However, substantive weaknesses persist in citation integrity, logical flow of mechanistic discussions, and clinical translatability of findings. These issues require resolution before the manuscript meets publication standards.
1. Citation Deficiencies: The most critical flaw is the inadequate support for key claims through essential references. While Sections 1 and 2.2-2.3 (trial data) are appropriately referenced, the remainder exhibits concerning citation sparsity. For instance, foundational statements in Section 2.1 regarding CTLA-4's role in regulating immune tolerance and preventing autoimmune reactivity lack supporting citations, as do claims that ipilimumab promotes T-cell activation and enhances cancer cell recognition. Furthermore, the citation of Swedish registry data (Ref. 70) in the Discussion fails to contextualize these findings within existing overall survival literature, diminishing its interpretive value.
2. Structural and Conceptual Weaknesses: Persistent organizational deficiencies undermine the manuscript's coherence, particularly concerning biomarker content (TMB, CTCs, PD-L1, ctDNA), which remains fragmented across Sections 2.5 (Resistance) and 4 (Biomarkers). Section 2.5 inadequately explains these biomarkers' roles as resistance mechanisms (if they have) and presents resistance biology without sufficient logical synthesis. Crucially, the mechanisms by which HDAC inhibitors and combination therapies overcome primary resistance to improve checkpoint inhibitor outcomes require explicit elucidation. Moreover, the distinct biological underpinnings and therapeutic implications of acquired resistance must be clearly differentiated and systematically explained, rather than being conflated with primary resistance mechanisms.
3. Content Organization and Terminology Issues: Section 2.6 should focus specifically on irAEs and toxicities in the adjuvant setting, including relevant comparisons to toxicities observed in advanced melanoma or other malignancies. The current version contains excessive tangential material. Consequently, Section 2.7's discussion of reproductive toxicity—being inherently related to toxicity—should be consolidated into Section 2.6 to maintain thematic focus on the adjuvant melanoma population. Both sections require rigorous editing to eliminate non-essential digressions. Additionally, the inconsistent use of "ICI" (Immune Checkpoint Inhibitor) and "CPI" (Checkpoint Inhibitor) terminology throughout the manuscript must be standardized to enhance clarity.
4. Discussion Lacks Clinical Translation: The Discussion insufficiently addresses the high Number Needed to Treat (NNT: 5–9 for stage IIB; 4–7 for stage IIC) to prevent one recurrence, without proposing concrete mitigation strategies (e.g., ctDNA-guided patient enrollment, risk-adapted treatment duration). While appropriately noting the delayed availability of KEYNOTE-054 overall survival data (expected 2027), the manuscript fails to explore the critical implications of this evidence gap for current clinical decision-making, thereby limiting the section's practical utility for practicing oncologists.
Reviewer 3 Report
Comments and Suggestions for Authors
-
Author Response
We note that Reviewer 3 did not submit any comments.